# Urban Features in Rural Landscape: A Case Study of the Municipality of Skawina

**Magdalena Wilkosz-Mamcarczyk** [1],*, **Barbara Olczak** [2]  and **Barbara Prus** [1]

1   Department of Spatial Management and Landscape Architecture, Faculty of Environmental Engineering and Land Surveying, University of Agriculture in Krakow, Balicka 253 c, 30-198 Kraków, Poland; b.prus@urk.edu.pl

2   Department of Ornamental Plants and Garden Art, Faculty of Biotechnology and Horticulture, University of Agriculture in Krakow, Aleja 29 Listopada 54, 31-425 Kraków, Poland; barbara.olczak@urk.edu.pl

*   Correspondence: magdalena.wilkosz-mamcarczyk@urk.edu.pl; Tel.: +48-12-662-40-17

**Abstract:** Events associated with growing suburbanisation and transport infrastructure lead to changes in the use of rural land reaching further and further from the central city. The present research focuses on the impact of the location of the municipality of Skawina and the functioning of its rural areas in the impact zone of Kraków. The first step to determine the transformations in the municipality of Skawina caused by the growth of Kraków and its suburban zone was to investigate the internal conditions in the municipality, such as its spatial development or functional structure, and external conditions such as its demographic transformations. Next, the development of the settlement structure in recent years and land management changes were investigated. The paper focuses on the introduction of typical urban features and functions into rural areas to identify the transformations and their dynamics. The status of the space was diagnosed by interpreting the map documentation for the administrative boundaries of the municipality of Skawina, regarding the presentation of the spatial development in 2006 and in 2014 to 2016, by verifying the existing planning documentation, and by visiting the site. The conclusions can be the basis for guidelines to protect the traditional rural and cultural landscape near Kraków.

**Keywords:** semiurbanisation; urbanisation; suburban development; suburban villages; traditional rural landscape

## 1. Introduction

Rapid economic development, population growth, and increasing industrialisation lead to a faster and more substantial urban expansion, which has had an impact on the historical environment of rural areas. Faced with social and economic changes, historical sites become endangered by neglect and deterioration or surrender to the pressures from new buildings or hardscaping. Without sufficient protection, many rural areas are affected by urban expansion, which is evident in new spatial conflicts, among other things. Human activities are significantly more destructive of tangible and intangible cultural heritage in rural areas than natural disasters [1–4].

Kamiński [5] categorises the consequences of urban expansion into rural areas as:

- spatial and environmental, meaning the expansion of developed environment at the expense of the natural environment and rural landscape, while adapting features of the urban landscape.
- sociocultural, particularly regarding the urban lifestyle of people living in villages undergoing urbanisation and urban behaviour in nature;

- economic, such as ongoing changes in the farming profession, increasing share of people with two professions in rural areas, and development of the multifunctional nature of the countryside;
- political, such as the growing importance of interest groups other than those focusing on agricultural production productivity.

The expansion of urban structures known as urban sprawl was noted in the USA in the 1960s. In the 1950s, the first efforts were made in the USA to study systematically the interrelationship between transport and the spatial development of cities. Deconcentrated urban land use was mostly associated with factors related to the general drift in developed societies and government policy regarding spatial planning. Urban changes stem from general changes in western societies such as increased wealth, economic and demographic changes, innovation in transport technology, and historical city development [6].

A study in China on the Tongzhou District, in Beijing, from 1972 to 2015, confirmed a significant expansion towards the suburbs and a loss of arable land (particularly fertile land). It was related to low construction costs in the areas and convenient access to transport. The authors suggested that the government should initiate appropriate land management procedures and add arable land quality surveys to the city's spatial planning system, including the monitoring of plans for new residential developments in rural areas [7].

Local governments often sell construction ground at low prices in order to encourage people to build and to increase the amount of taxes. The consequences of this are apparent in many towns in Europe, where residential areas of single-family dwellings are growing close to the towns and nearby villages. This competition between municipalities is likely to continue [8].

Cultural heritage in rural areas does not consist only of intangible heritage, but also, and above all, of development layouts, architecture typical of local communities, and land arrangement. In many countries (such as Norway), agriculture is seen both as a threat (often large-area, mechanised agriculture) to, and a caretaker (mostly traditional crops) of, cultural heritage [9]. Cultural heritage is the sum of two components. The first one is what nature provides, such as areas with valuable environment or used for farming. The other is the anthropogenic contribution, in the form of settlement arrangements or traditional development typical of a region and determined by the local lifestyle [10]. By extracting and recognising cultural heritage landscape, it is possible to reflect regional cultural characteristics. Traditional villages are the most representative kind of regional cultural heritage landscape, embodying and preserving the traditional culture [11]. From a historical perspective, typical distances between two localities in many European countries range from 3 to 5 km because the land between them was needed for agricultural production, to feed the people in the villages. Nowadays, these villages and towns are often entwined [8]; agricultural locations have increasingly more characteristic urban components [12,13].

The landscape of the Polish countryside has been changing significantly since the last decade of the 20th century. These changes have been caused by functional transformations, the political and economic transformation of 1989, or the contribution of European funds [14]. The growing globalisation and suburbanisation have contributed to the dynamic development of suburban areas as well, which led to the transformation of rural areas [8,15,16]. The rapid advancement of transport and urbanised areas resulted in regions that are neither urban nor rural in the traditional sense [17].

The interweaving of the urban structure and lifestyle with rural areas is most evident in Poland in the impact zones of the largest cities. These changes are further promoted by the slowdown in rural development, rural-to-urban and urban-to-rural migration, and the increase of non-agricultural activities in the countryside. They are conditioned by the semiurbanisation of rural settlements, which is defined as the socioeconomic and morphological transformations of the countryside resulting in its partial but not complete urbanisation. Rural areas are not incorporated into city administration limits; neither do they turn into a developed city [18]. These changes are apparent in changed rural spatial management, architecture, society, and landscape [19]. Spatial conflicts are a common phenomenon in places where the traditional regional layout of the village is penetrated by modern

residential developments of urban nature, followed by an urban lifestyle at variance with the traditional regional culture.

The current condition of agriculture in Poland was determined mostly by historical events that affected the ownership situation, political changes, and military conflicts. The Polish agrarian policy had been shaped by the policy of the respective invader since 1918 (Austria, Prussia, and Russia). In the territory of what is today Małopolskie Voivodeship (where the region of Skawina is located), inheritance was governed by the Napoleonic Code. Land was inherited equally by all children who could not abandon the agricultural holding to find non-agricultural jobs. It was then that the Polish notion of a proverbial "Galician poverty" was coined. This resulted in a significant fragmentation of agricultural holdings [20]. The land use trends in the area have changed after World War II. The agrarian reform of 1944–1945 abolished feudal demesne farm ownership. The government promoted large-area peasant holdings (Polish kulaks). As a result of the reform, 1953–1955 saw four types of ownership: individual holdings, state agricultural holdings, agricultural production cooperatives, and farmer clubs [21]. The sociopolitical transformation of the 1990s affected the form and functioning of agricultural areas to a great extent. It resulted in the transformation from state agricultural holdings to private ownership and the possibility to purchase buildable land by citizens. In this way, Poles, who often lived in blocks of flats, could pursue their dream of owning a house.

However, the number of parcels available for single-family development within the cities' boundaries was limited. Moreover, their prices were much higher than in suburban areas. This is why the Polish cities of the post-socialist era started to sprawl into open areas. New developments can be found mostly at the interface of urbanised and open areas and along main transport routes. Suburban villages became more urbanised as well.

It was not possible to make long-term and consistent plans for rural areas because of the many politically motivated agrarian reforms in Poland. There were nine major agrarian reforms in the period of 60 years from 1933 to 1999. Political transformations towards a free-market economy after 1989 were reflected in exceptionally dynamic changes in the spatial and functional structure of rural settlements. As a consequence, the social and economic structure of the then-agricultural village changed. Rural urbanisation was mostly presented, at the time, as an overwhelming process of change, aimed at replacing old structures. The idea of urbanisation was mostly a negation of rurality understood as a particular cultural tradition and a sense of identity, social, cultural, economic, and landscape otherness [18]. The modern rural layout has come to be considered a vitally important component of the heritage—and thus of the local resources—of the village. It is defined by law as a "rural planning scheme with building complexes, individual buildings, and landscaped green areas arranged according to historical ownership and functional structure, including that of streets or roads." [22].

The suburbs have developed differently in different countries. Many researchers [23–25] have confirmed that the trends in post-communist countries are similar to those in developed, capitalist countries, but slower due to the nature of their socioeconomic growth. The countries have specific historical factors, geographical conditions, housing stock, economic processes, and urban-to-rural migration [26,27]. Because of the centralised socialist planning, urbanisation in Poland had a strikingly different nature than in capitalist cities. Most urban regions here were monocentric: from the countryside (the provider for the city) towards the core of the city, where jobs and services were concentrated. The sociopolitical transformation introduced a market economy. This socioeconomic reorganisation has been reflected in the transformation of the landscape of former socialist cities and their surroundings. Suburbanisation has become the dominant mode of city growth, and with it spatial dispersion. New commercial and industrial zones were established as it progressed. New project schemes paved the way for new suburban job hubs [27]. The suburban growth phase was bound up with the size of the city and the level of economic development. The influx of the urban population to the suburbs started in the centres that completed the transitional period [28]. Compact cities sprawled outside urbanised areas, and outside the cities' limits. The growth of housing takes place mostly at the interface of urbanised areas and suburban villages. The urban and rural ideas

of space and lifestyle clash, leading to spatial chaos, degradation of the cultural and natural landscape of rural areas, and social conflicts. The present research aims to diagnose these spaces in order to preserve the cultural heritage of the Lesser Poland village. In our view, the first step is to determine and understand all the conditions related to this process.

Suburban municipalities near cities have been undergoing multidimensional changes in recent years. Spatial transformations are related to the demographic structure and spatial management, including the type of development, land use, and infrastructure. The processes in these areas are triggered by the proximity to the city [29]. Researchers have demonstrated, among other things, that the agricultural function is dwindling in these areas [30].

The urbanisation of the Polish village was in line with the post-war ideology of social transformations. Affected by the communist ideology and development policy, it depicted the rural culture as outdated and failing to follow transformations towards an urban-industrial society [31]. The housing landscape of large suburban agglomerations and along main transport routes (motorways) has changed, often into residential and commercial complexes [32]. Today, the rural areas near most large cities are socially, architecturally, and functionally diversified. It is often disorganised and spatially chaotic. Architectural styles and planning trends are often interwoven, mostly due to administrative decisions. The urban culture has replaced the rural one, and its spatial range keeps growing [33]. The developments in suburban settlements are much less dense. They have lower buildings and larger parcels, but the trend is towards smaller parcels with more compact building arrangements.

Most rural buildings have only one storey, and often a basement. The construction material is brick or lightweight concrete blocks. The roof is made of roof tiles or steel roof tiles. The nature of developments exhibits significant changes as well. The development fabric is punctuated with typical urban structures (ready-made, universal house plans). The number of storeys and the volume of new buildings are growing. The villages can be considered multifunctional, with only part of the residents working as farmers. As the function of the village changes, residential and farming building complexes disappear.

The spatial form of the Polish countryside has been shaped not only by environmental changes but also by cultural and historical factors. Its landscape includes religious and traditional components but also vegetation, adding to the diversity of the system. The origins of the building arrangement of the countryside date back to the Middle Ages. Their focal point could be a public square—for example. in the case of a roundling or a road resulting in a linear settlement. Buildings characteristic of most Polish villages include a church (in larger villages), wayside chapels and crosses, manors, and manor farm buildings. Peasant buildings in the Kraków countryside consisted mostly of a low house near the road and farm buildings. In front of the house, there was the entrance garden with typical rural flowers: hollyhocks (Alcea sp.), paeonies (Paeonia sp.), and cosmos (Cosmos sp.). Behind the house there was an orchard and a strip of agricultural land.

Farming was not a monoculture. It usually catered for the family, and the excess produce was sold. The 20th century brought significant changes to Polish agriculture. Potatoes, rye, and oat lost their significance, and the share of wheat, rapeseed, barley, and corn increased. Farmers appreciated triticale. They kept cattle, swine, and poultry. Today, the structure grows increasingly hazy. Husbandry is diminishing, while non-agricultural activities grow more popular. Some holdings are transformed into large-area farms (following the model of monoculture, mechanised farms). The first schools appeared in rural areas between the 19th and 20th century. Convenience stores and bars (inns) were scarce and usually located in private houses. Today, most villages have a (primary) school and a grocery store.

Nowadays, rural areas in urban impact zones are endangered by the migration of urban development features and functions. As transport infrastructure develops, villages far from the central city are susceptible to the adverse consequences of such transformations. Suburban villages and small towns within commuting range to medium and large cities go through a transformation of rural or small-town development features into typically urban ones [34].

Multiple visions and strategies for rural development have been proposed in Poland for the national and voivodeship level. They focus on such matters as the formation of agricultural production in line with environmental requirements and in such a way as to preserve landscape qualities. They are, however, too general or even invalid. There is still room for more in-depth documents because the processes are very dynamic [32].

The objective of this paper is to analyse spatial and planning trends identified in modern villages located in the impact zone of a large city, using the municipality of Skawina as an example. The first step to determine the transformations in the municipality of Skawina caused by the growth of Kraków and its suburban zone was to investigate the internal conditions in the municipality, including its spatial development or functional structure. Among the external conditions, demographic transformations deserved particular attention. Afterwards, the development of the settlement structure in recent years and land management changes were investigated. The analysis involves a mixed urban and rural municipality administrated from the town of Skawina, classified as a medium-sized town, located between Kraków and the rural zone of the municipality. The primary objective of the paper was to analyse typical urban features and functions introduced into rural areas, and to identify the transformations and dynamics of the changes. The data can be used to build schemes to prevent the adverse effects of the semiurbanisation of rural areas in the municipality of Skawina.

The paper focuses primarily on spatial issues and changes in the nature of developments, as they are undoubtedly a result of processes related to suburbanisation and indicate the severity of these problems.

Research on modern spatial phenomena pays much attention to large cities [35]. Towns are not such a popular topic, which is why Skawina was selected for the present research.

Skawina is situated in the suburban zone of Kraków (15 km from Kraków's city centre). This fact entails a rapid growth of the settlement structure within the municipality and the majority of its population commuting to Kraków. The analysis involves a mixed urban and rural municipality administrated from the town of Skawina, classified as a medium-sized town, located between Kraków and the rural zone of the municipality. The research involved the following villages: Borek Szlachecki (BS), Facimiech (FA), Gołuchowice (GO), Grabie (GR), Jaśkowice (JA), Jurczyce (JU), Kopanka (KO), Krzęcin (KR), Ochodza (OC), Polanka Hallera (PH), Pozowice (PO), Radziszów (RA), Rzozów (RZ), Wielkie Drogi (WD), Wola Radziszowska (WR), and Zelczyna (ZE). Being an industrial satellite town of Kraków, Skawina is growing more important thanks to its thriving industry and transport network (the A4 motorway), which attracts new residents. It is considered an alternative settlement location for residents of Kraków because of its small-town character, contrasting with the large-city atmosphere of the second-largest city in Poland, Kraków. The villages in the municipality have a traditional rural landscape, which invites new residents who seek calm locations not far from a large city.

The municipality covers an area of 9984 ha and has 43,496 residents (at the end of 2018) [36]. The town of Skawina covers 2050 ha and has 24,325 residents. The rural area of the municipality consists of sixteen villages across 7934 ha with 19,171 residents. To the north, it borders on Czernichów, Liszki, and Kraków, and to the east on Mogilany. To the south of the municipality, we find Myślenicki District, with the Myślenice and Sułkowice municipalities, and to the west Wadowicki District, with the municipalities of Lanckorona, Kalwaria Zebrzydowska, and Brzeźnica [36 Główny Urząd Statystyczny].

An important factor affecting the economic growth of the town of Skawina was the establishment of an aluminium smelter after World War II. Other investments followed: a power plant and new housing developments. The population grew, as the nearby plants promised job opportunities. The trend continues today. Agriculture is slowly dying; arable land is often set aside or transformed into fallow land [37]. Residents have mostly abandoned agriculture in favour of jobs offered by numerous businesses in the area.

Spatial transformations are directly related to migration. Miasta województwa małopolskiego—zmiany, wyzwania i perspektywy rozwoju [Cities in the Małopolskie Voivodeship—Changes, Challenges, and Perspectives for Growth] offers a typology of municipalities in line with their current phase of

suburbanisation (strong moderate, initial), based on net migration, net urban-rural and rural-urban migration, and rural inflow. Skawina was among the places with very advanced suburbanisation processes from 2014 to 2016 in the Małopolskie Voivodeship [38].

## 2. Materials and Methods

### 2.1. Research Method

Suburbanisation is manifested not only through urban features appearing in the rural landscape but also in a broader spatial context. This is why the research was divided into three stages and problem perspectives (Figure 1).

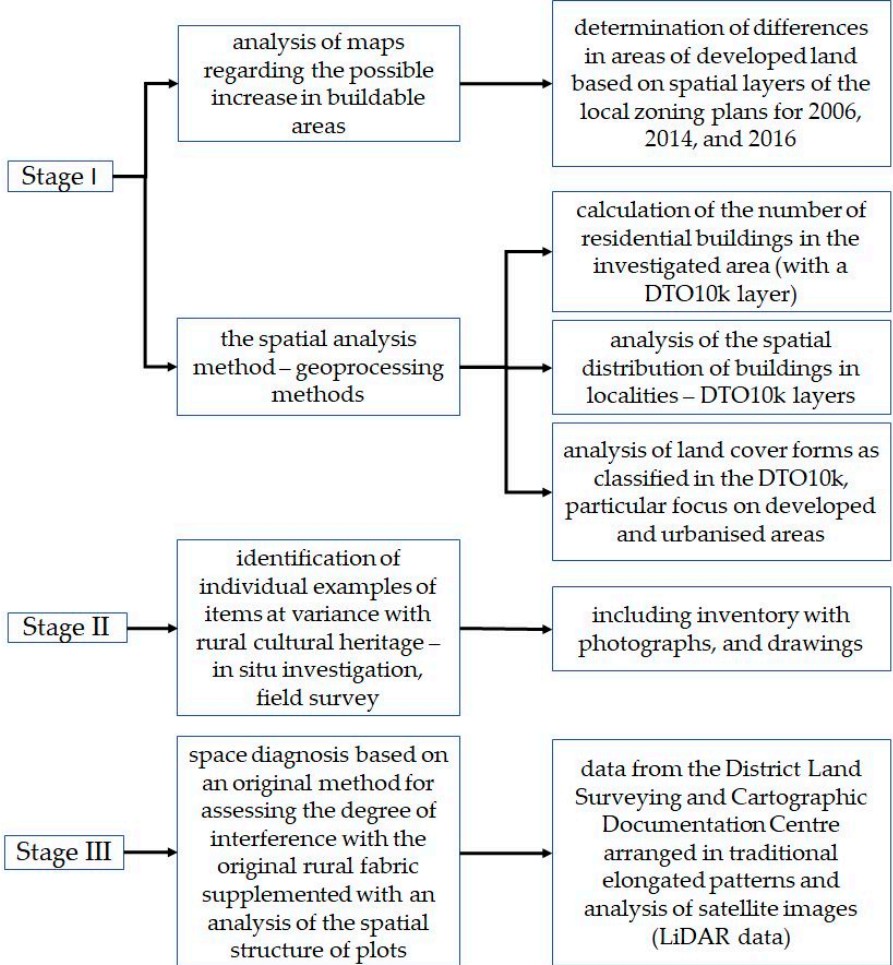

**Figure 1.** Research method diagram.

**Stage 1** involved the analysis of maps to identify the area earmarked for development and its changes in a broader temporal perspective. It was achieved by comparing drawings in local zoning plans for 2006, 2014, and 2016. Several spatial analyses were performed at this stage. Among them, the number and spatial distribution of residential buildings in the area or types of land cover were determined. Data for the analysis were obtained from the Database of Topographic Objects, DTO10k. The database provides detailed and accurate spatial data from 1:10,000 topographic maps. It contains information about the spatial location of topographic objects, their descriptive attributes, and such domains as the water network, transport network, land cover, or means of environmental protection [39]. The purpose of the DTO10k is to collect and provide data that can be used in various information systems [40].

**Stage 2** consisted of the identification of specific examples of components that were at variance with the rural cultural heritage. It was based on field surveys and a comparative analysis of local zoning plan documentation for 2006, 2014, and 2016.

The years 2006, 2014, and 2016 were when local zoning plans were adopted for the town and municipality (2006), the municipality (2014), and the town (2016). The plans were analysed in depth for changes in areas where building is planned. Land earmarked for development in the plans for 2014 and 2016 was characterised and compared to data in the plan for 2006. Planning documents are prepared and adopted under statutory procedures in Poland. They can, therefore, be implemented in intervals of several years.

**Stage 3** was a diagnosis of the space based on an original method for assessing the degree of interference with the original rural fabric. It adopted a numerical taxonomy method with a synthetic taxonomic metric of development.

The major identified settlement network transformations related to demographic changes in the municipality of Skawina included intensified development with new single-family houses of urban nature. The analysis covered features alien to the rural landscape, including street furniture, hardscape, façade colours contrasting with the landscape, modern gardens, forms of buildings at variance with rural traditions, and modular fences. All the listed elements were covered by the field survey. The original values were noted as a percentage of alien forms in the total number of buildings in the localities. The normalised value was calculated by expressing a variable as a multiplicity of the mean, using Formula (1):

$$x_i' = \frac{x_i}{x_{av}} \tag{1}$$

where: $x_i'$ is the normalised value;

$x_i$ is the value calculated based on the field survey;

$x_{av}$ is the mean value of variable $x$.

Linear ordering methods are a subgroup of multidimensional comparative analysis methods. There are many algorithms available for calculating synthetic metrics based on selected diagnostic variables. The first to propose a synthetic metric of development was Z. Hellwig [41]. His metric compared the economic development of selected countries. Methodologies for building taxonomic metrics for various applications have been developed by Bartosiewicz (1976) [42], Strahl (1978) [43], Zeliaś and Malina (1997) [44], Kukuła (1986, 2000) [45,46], Walesiak (2003) [47], Gatnar and Walesiak (2004) [48], and Tarczyński and Łuniewska (2006) [49]. In the present paper, the synthetic index normalised values of variables were calculated based on Formula (2) [41]:

$$w_s = \sum_{i=1}^{5} x_i' \tag{2}$$

According to the assumptions of numerical taxonomy, the calculated synthetic index represents an aggregate outlook on changes in the rural landscape caused by the introduction of features at variance with the traditional landscape in each investigated village in the municipality by combining five categories of investigated features.

The units (localities) were classified using the Jenks natural breaks method. Four classes were identified based on the values of the synthetic index. High values of the index reflect a significant transformation/occurrence of alien forms in the landscape, while low values mean a low level of transformation in the units (localities).

The investigation was supplemented with an analysis of the spatial structure of the parcels. This stage involved a spatial analysis of data from the District Land Surveying and Cartographic Documentation Centre for parcels arranged in traditional elongated patterns and the analysis of satellite images (LiDAR data).

## 2.2. The Investigated Area and Local Conditions

The investigated area was preliminarily identified using LiDAR images representing changes in the land cover structure of the suburbs of the city and the town. The research encompassed the boundaries of Skawina, in the Małopolskie Voivodeship, bordering on Kraków (Figure 2a,b). The municipality is part of the Kraków Metropolitan Area (KMA) [50]. Municipalities in Poland are classified into three administrative categories: urban municipalities (with boundaries identical with the limits of the town or city that is the municipality), mixed urban and rural municipalities (that include both a town or city and land outside of its limits), and rural municipalities that do not have a town or city within their boundaries [51].

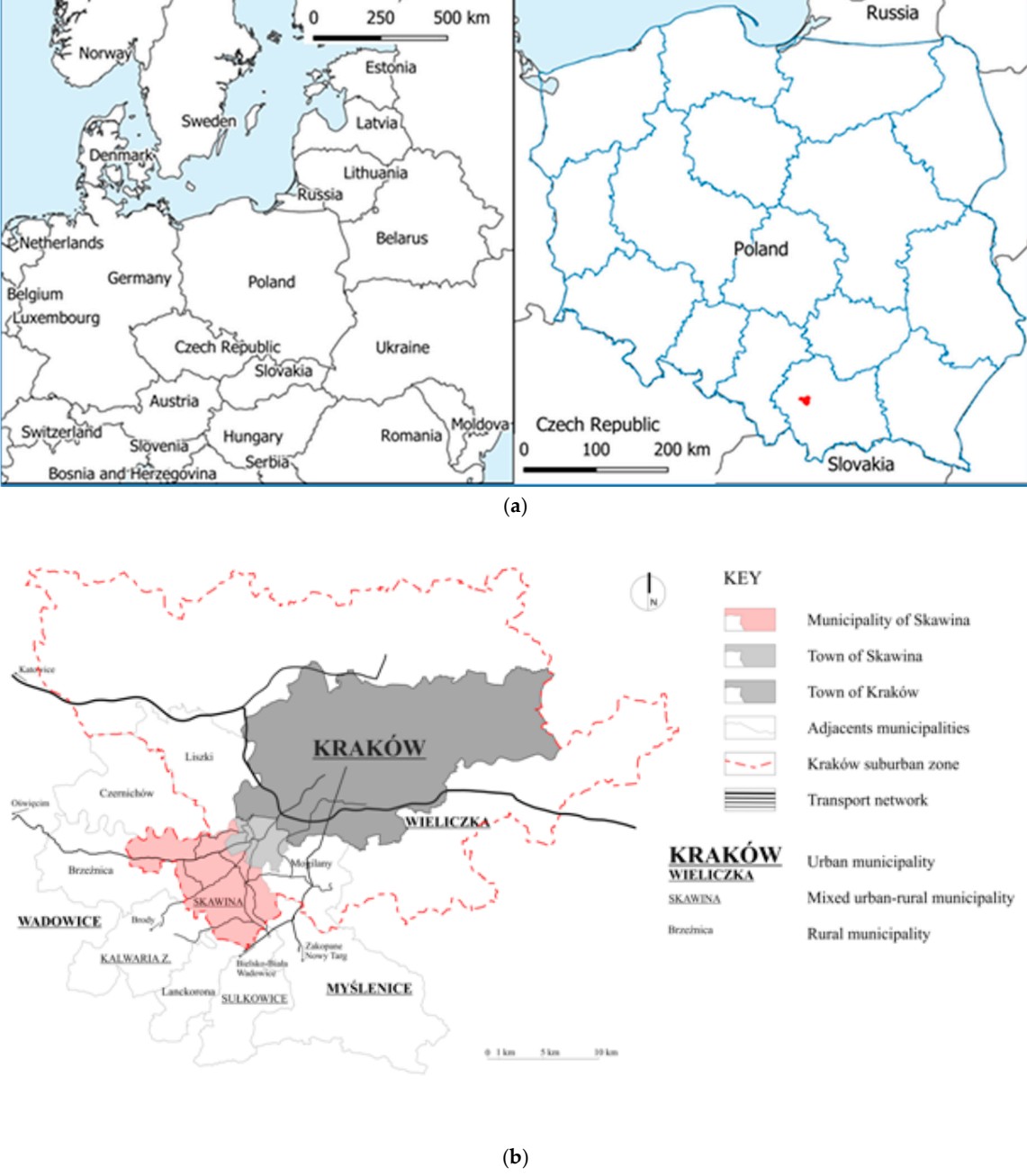

(**a**)

(**b**)

**Figure 2.** Relations of Kraków and its suburban zone. (**a**) The investigated area, the municipality of Skawina (**b**) Municipality of Skawina, the city of Kraków, and its suburban zone, based on Amended.

The largest increase in the number of new apartments and population size in the Kraków Metropolitan Area between 1989 and 2002 was recorded for the municipality of Skawina, among other places [50]. Studies show that the number of building permits issued in the 1990s in municipalities around Kraków was 1.5 to 4 times higher than for land within city limits. Note that the permits were granted mostly to former residents of Kraków, which demonstrates a great interest in suburban areas among this group [50].

## 3. Results

### 3.1. Stage I. External And Internal Conditions Location in the Impact Zone of a City (Kraków)

Rakowski [52] and Prochownik W. [53] attempted to determine the characteristics of the urbanisation of the Polish countryside. The latter differentiated between urbanisation and deruralisation of the countryside, which included agricultural modernisation and reflected the extinction of the traditional countryside. The study also focused on the transformation of the spatial structure of rural settlements (spatial management), such as the increased complexity of settlements and transport systems [33] and non-agricultural activities.

The paper presents development trends identified in modern rural areas located in the impact zone of large cities in Poland. Note the introduction of features of the Sub-Tatra architectural style among traditional rural buildings and the modern residential building with a flat roof and plans for a large amount of glass (as shown in the photographs).

An analysis of environmental and spatial consequences caused by the expansion of urban zones into rural areas [5] and an analysis of the changes in the intended use and management of land in the 2006 and 2016 local zoning plans for the municipality indicated that the area of land earmarked for housing developments has been growing for ten years (Figures 3 and 4). The increase in built and urbanised areas in the land cover structure is apparent when comparing local zoning plans. The areas of single-family housing, multi-unit residential housing, and industrial zones have grown (see Figure 3). The increase in areas of new projects and schemes was at the expense of agricultural land and grassland (Figure 4).

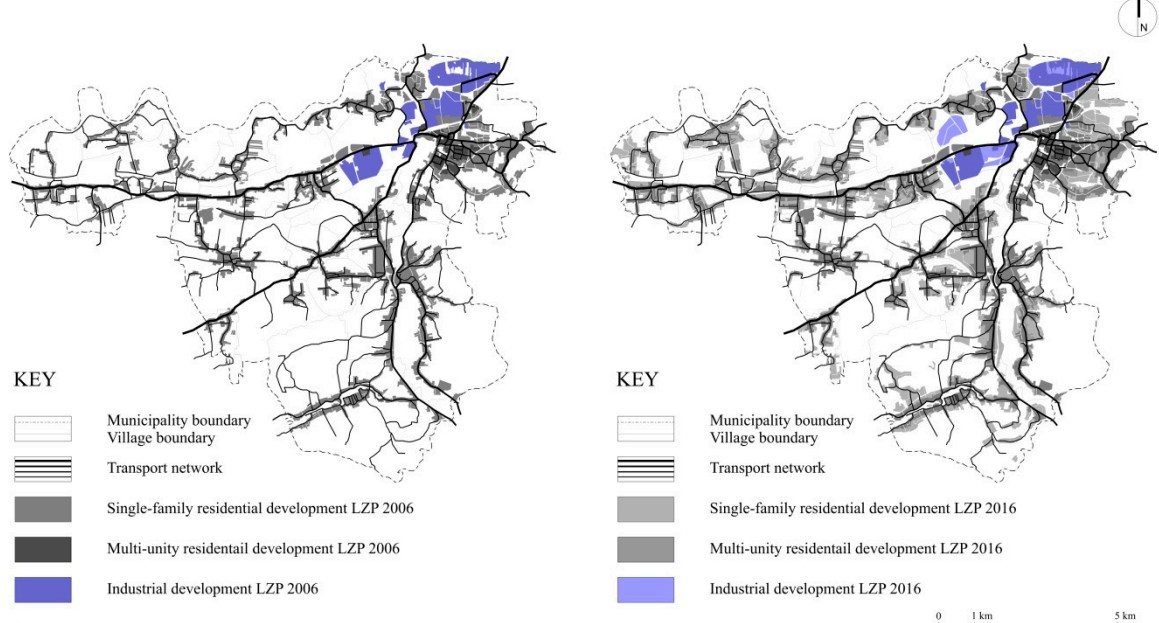

**Figure 3.** Changes in the areas of land to be developed in the municipality of Skawina in 2006 and 2016 in the local zoning plans, based on the local zoning plans for 2006, 2014, and 2016.

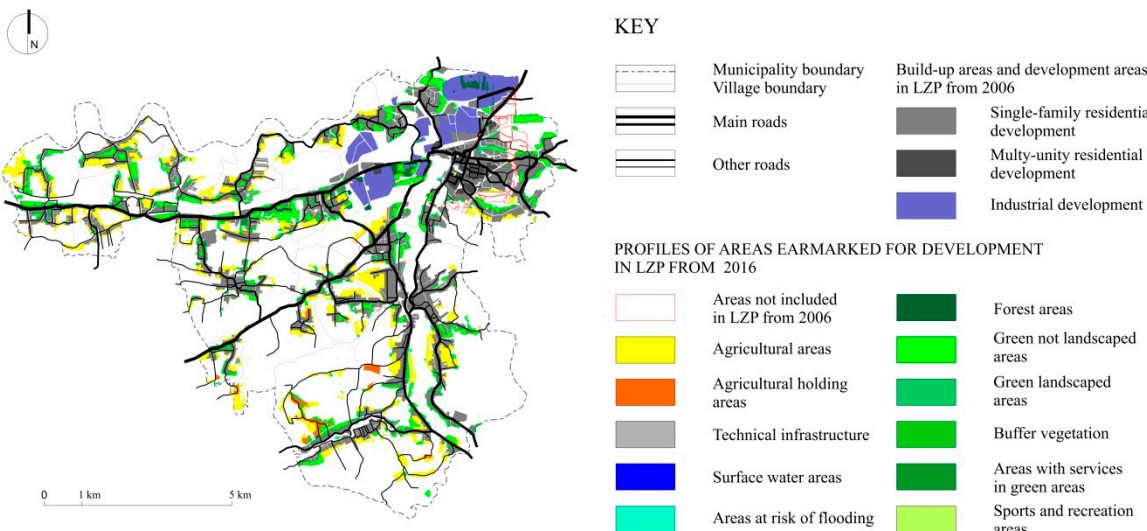

**Figure 4.** Specification of buildable land in the 2016 local zoning plan for the municipality of Skawina and in the 2014 local zoning plan for the town, based on local zoning plans for 2006, 2014, and 2016.

Demographic transformations are one of the key drivers of settlement structure changes. The population of the municipality of Skawina grew by 2.8% from 2009 to 2014, but the dynamic has been dwindling since 2013. Most of the population, 56.1%, live in the urban area. At the same time, the rural areas have exhibited a greater dynamic of population growth [54]. An analysis of the pace of population changes provides the first clue to identifying areas where suburbanisation transformations can be expected. According to the Local Data Bank [36], the population change dynamic for the period from 2010 to 2016 indicates a 2.51% to 5% positive growth in Skawina. The growth of population directly increases the population density in the same period, resulting in 250 to 350 people per km$^2$ in the municipality of Skawina in 2016 [38]. This trend results in more buildings and an increase in developed areas. The municipality has over 9.3 thousand buildings (in addition to the town of Skawina), including over 5.1 thousand single-family houses ( Figure 5; Figure 6).

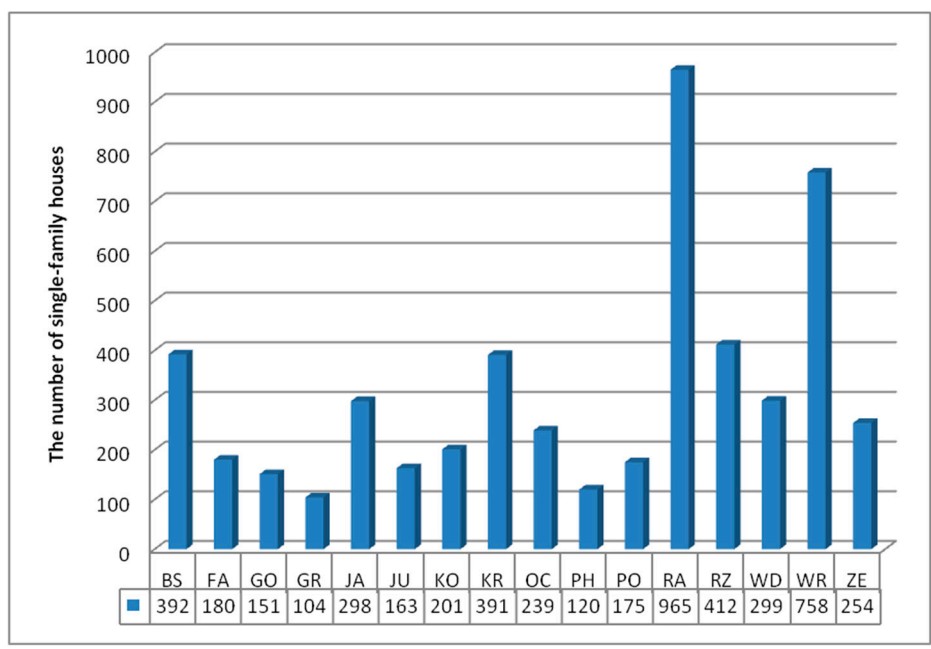

**Figure 5.** The number of single-family houses in the analysed districts of the municipality of Skawina.

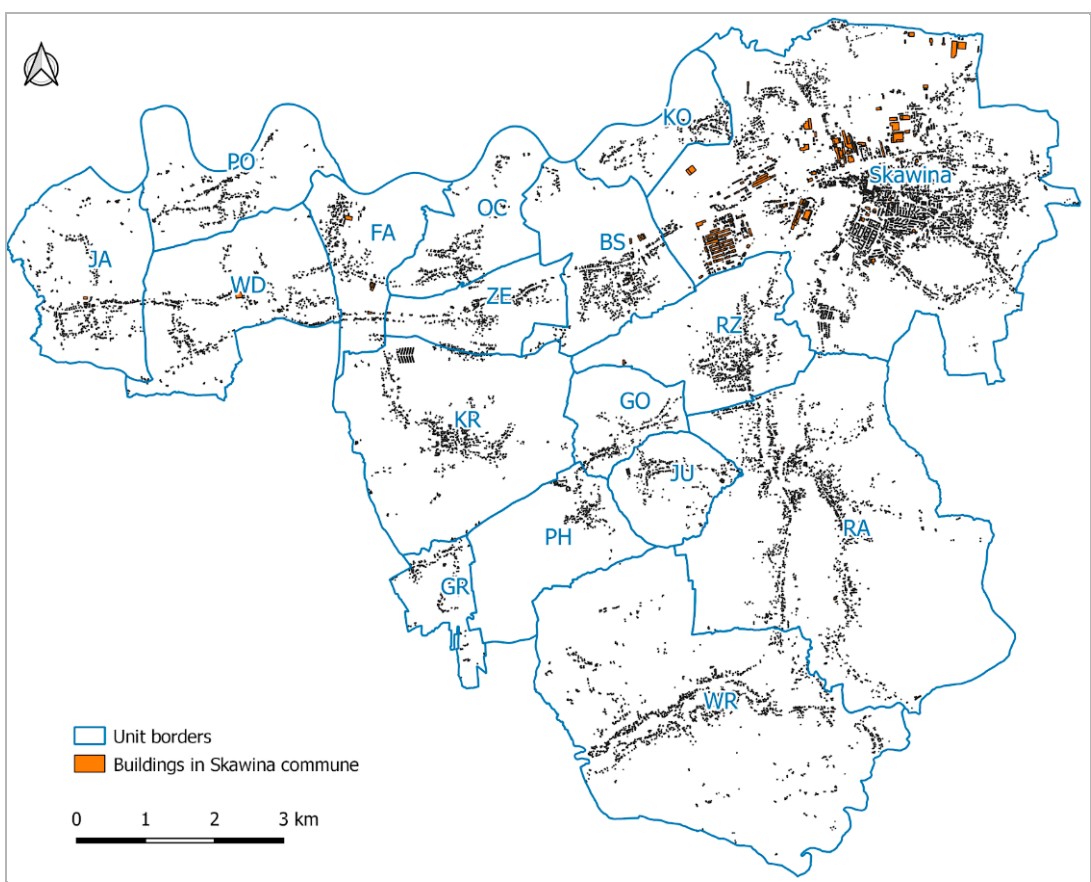

**Figure 6.** Distribution of buildings in localities in the municipality of Skawina according to the DTO10k (Database of Topographic Objects) in 2019.

Suburbanisation changes in the investigated area are evident in the land-use structure as well. Developed and urbanised land constitutes 13.58% of the municipality. The type of municipality affects its functional structure. About 28.49% of the area of Skawina is covered by arable land. Forests cover about 17% of the area, and grassland 36.5%. Surface water can be found on 1.6% of the total municipality area. Orchards occupy 1.37%, roads 0.94%, and wasteland 0.09% (Figure 7).

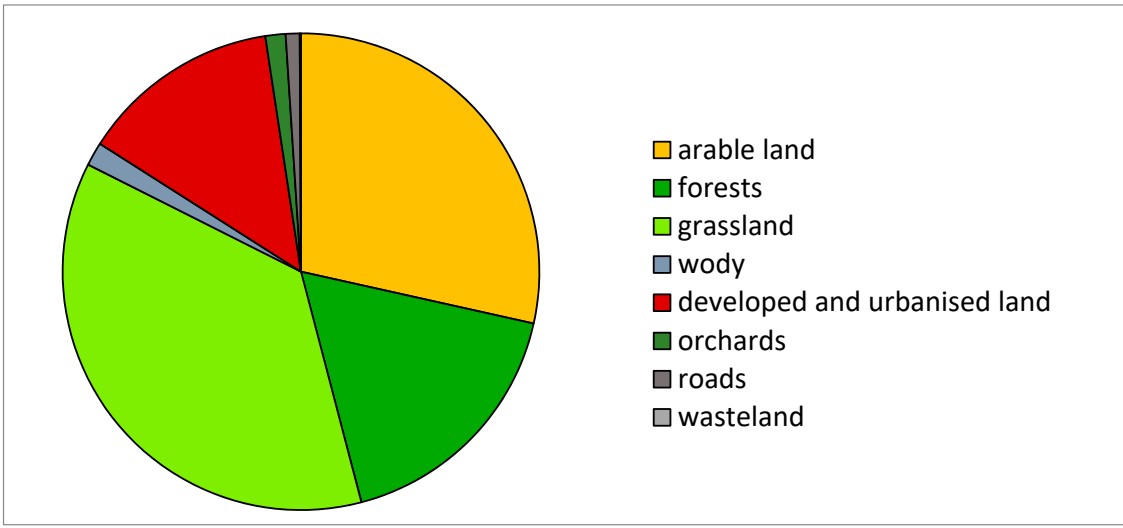

**Figure 7.** The percentage share of land cover types in the total area of the municipality of Skawina.

The non-urban landscape of the municipality has been shaped by the dominant agricultural function and the rural land management approach it entails [DTO10k Figure 8].

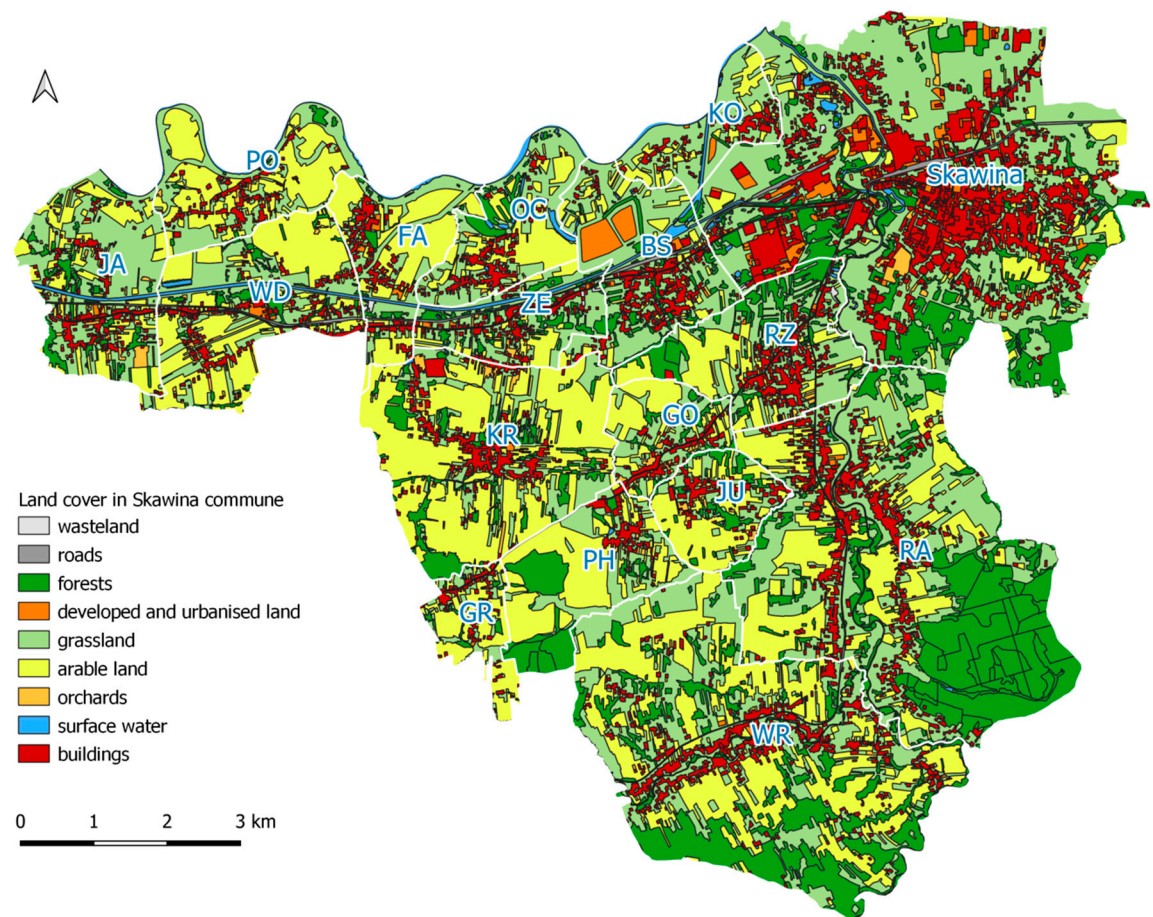

**Figure 8.** Land cover in districts of the municipality of Skawina according to the BDOT10k.

### 3.2. Stage II. Identification of Individual Examples

Based on developable land in the local zoning plans and in-situ analyses, changes in the building profile, the introduction of urban and alien features into rural areas, and the decrease of the agricultural function were noted as adverse spatial and environmental phenomena. The particular focus was on the dissonance caused by the heights of buildings up to several meters and on forms at variance with the traditional architecture of the Polish countryside. New buildings resemble miniature castles or manors. Another popular component is modular fencing. Walls or precast concrete panels with wooden texture are the go-to solutions in many villages [55]. The same is true for the surroundings of residential buildings. Gardens that used to be full of diverse trees, shrubs, and flowers were turned into grassy surfaces surrounded by "thuja walls". As regards the sociocultural domain, types of behaviour and lifestyles considered "urban" and conflicts with original residents of suburban areas and migrants are increasingly common. The conflicts may stem from field operations carried out often until late in the evening and odours and sounds produced by animals (Figure 9).

Land management components typical of agricultural land use, the rural profile of the settlement network, topography, and land cover constitute the spatial system typical of non-urban areas, which has to be protected with planning documentation to preserve rural cultural values (Figure 10).

The occurrence in the rural landscape of alien urban elements borrowed from another function, settlement, or scale causes spatial dissonance. In large doses, it may result in spatial chaos and degradation of the rural cultural landscape and traditional regional values [15].

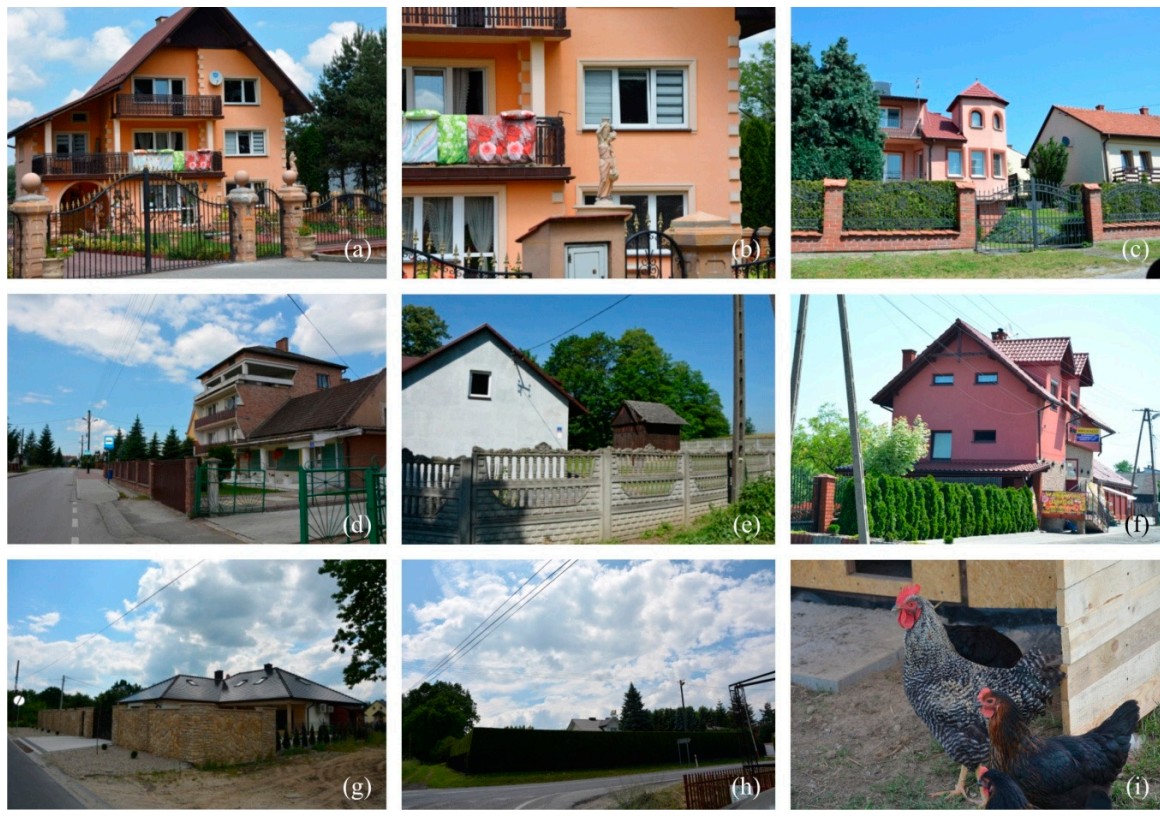

**Figure 9.** Processes related to the progressing semiurbanisation identified in the municipality of Skawina, part 2, (**a**)–(**c**).—Colour schemes of façades and introduction of exotic features, as identified in Wielkie Drogi and Polanka Hallera (**d**)–(**f**).—Construction dissonance as seen in Kopanka, Sosnowice, and Krzęcin, (**g**)–(**i**).—Discontinuity of relationships between rural residents and migrants from the city.

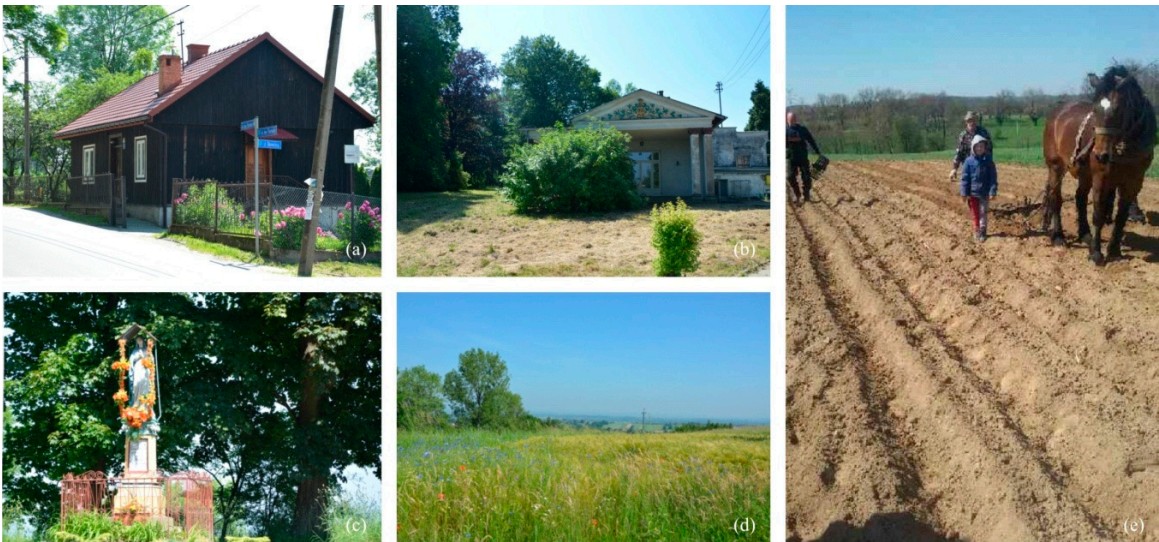

**Figure 10.** The countryside falling into oblivion. (**a**) Krzęcin, (**b**) Haller's Manor in Polanka Hallera, (**c**) Polanka Hallera, (**d**) A view of Krzęcin from Polanka Hallera, (**e**) Field operations.

### 3.3. Stage III. Determination of the Degree Of Changes in the Rural Landscape

Changes in the rural landscape in the municipality of Skawina (Table 1) calculated with the synthetic index are manifested in most localities, with a slight advantage of urban features such as building colour schemes and alien landscape features.

**Table 1.** Results of the field survey—investigation of features alien to the rural landscape.

| | | No. of Single-Family Houses | Street Furniture/Hardscape Alien to Countryside | | Façade Colours Contrasting with Landscape | | Modern Gardens, Typical of Terraced Housing: Lawn + Hedge | | Building form at Variance with Rural Tradition | | Modular, non-Traditional Fencing | | Synthetic Index |
|---|---|---|---|---|---|---|---|---|---|---|---|---|---|
| | | | V | NV | V | NV | V | NV | V | NV | V | NV | V |
| 1 | BS | 392 | 0.01 | 0.4 | 0.03 | 0.6 | 0.3 | 1.6 | 0.3 | 1.1 | 0.3 | 2.1 | 5.7 |
| 2 | FA | 180 | 0.01 | 0.4 | 0.05 | 0.9 | 0.4 | 2.2 | 0.4 | 1.4 | 0.03 | 0.2 | 5.1 |
| 3 | GO | 151 | 0.01 | 0.4 | 0.5 | 0.9 | 0.1 | 0.5 | 0.3 | 1.1 | 0.1 | 0.7 | 3.6 |
| 4 | GR | 104 | 0.01 | 0.4 | 0.05 | 0.9 | 0.1 | 0.5 | 0.4 | 1.4 | 0.05 | 0.4 | 3.6 |
| 5 | JA | 298 | 0.01 | 0.4 | 0.01 | 0.2 | 0.5 | 0.3 | 0.1 | 0.4 | 0.05 | 0.4 | 1.5 |
| 6 | JU | 163 | 0.01 | 0.4 | 0.03 | 0.6 | 0.2 | 1.1 | 0.3 | 1.1 | 0.01 | 0.1 | 3.1 |
| 7 | KO | 201 | 0.1 | 3.6 | 0.1 | 1.8 | 0.3 | 1.6 | 0.7 | 2.5 | 0.4 | 2.8 | 12.4 |
| 8 | KR | 391 | 0.3 | 1.1 | 0.1 | 1.8 | 0.3 | 1.6 | 0.2 | 0.7 | 0.1 | 0.7 | 6.0 |
| 9 | OC | 239 | 0.01 | 0.4 | 0.05 | 0.9 | 0.1 | 0.5 | Less than 0.1 | 0.2 | 0.02 | 0.1 | 2.1 |
| 10 | PH | 120 | 0.01 | 0.4 | 0.01 | 0.2 | Less than 0.1 | 0.3 | Less than 0.1 | 0.2 | 0.02 | 0.1 | 1.1 |
| 11 | PO | 175 | 0.05 | 1.8 | 0.03 | 0.6 | 0.05 | 0.3 | 0.2 | 0.7 | 0.8 | 0.6 | 3.9 |
| 12 | RA | 965 | 0.03 | 1.1 | 0.6 | 1.1 | 0.1 | 0.5 | 0.4 | 1.4 | 0.1 | 0.7 | 4.9 |
| 13 | RZ | 412 | 0.03 | 1.1 | 0.05 | 0.9 | 0.3 | 1.6 | 0.3 | 1.1 | 0.4 | 2.8 | 7.5 |
| 14 | WD | 299 | 0.05 | 1.8 | 0.1 | 1.8 | 0.3 | 1.6 | 0.4 | 1.4 | 0.4 | 2.8 | 9.5 |
| 15 | WR | 758 | 0.02 | 0.7 | 0.05 | 0.9 | 0.1 | 0.5 | 0.2 | 0.7 | 0.1 | 0.7 | 3.6 |
| 16 | ZE | 254 | 0.05 | 1.8 | 0.1 | 1.8 | 0.2 | 1.1 | 0.2 | 0.7 | 0.1 | 0.7 | 6.2 |

V—Value; NV—Normalised Value.

The values in Table 1 present the results of the field survey. Three of the investigated localities have a slightly greater share of changes. They occupy land directly adjacent to the town of Skawina, KO and RZ, and areas directly adjacent to the regional road in the village, WD (Figure 11; Figure 12).

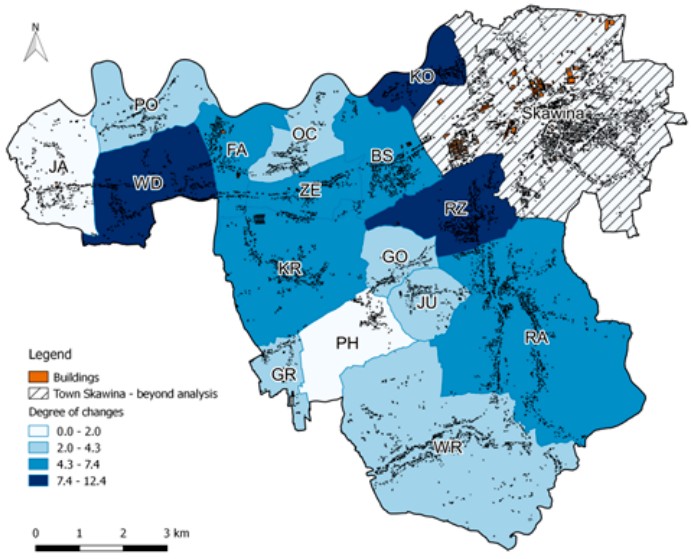

**Figure 11.** The degree of changes—alien features in the rural landscape of villages in Skawina.

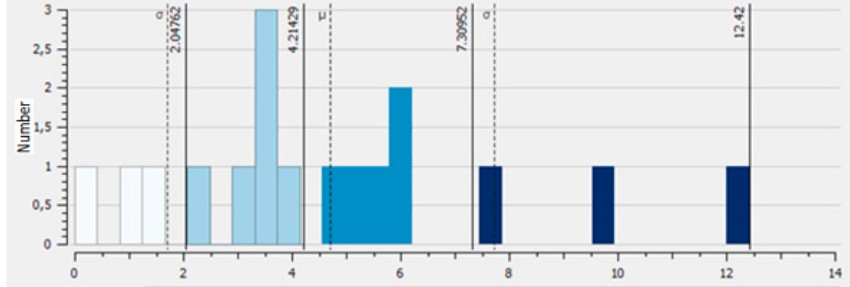

**Figure 12.** The dispersion graph of values for the degree of changes in the urbanisation phase in localities in Skawina.

The major identified settlement network transformations related to demographic changes in the municipality of Skawina included intensified development with new single-family houses of urban nature and the sizes and ratios of plot boundaries resulting from the elongated arrangement of agricultural fields (Figure 13).

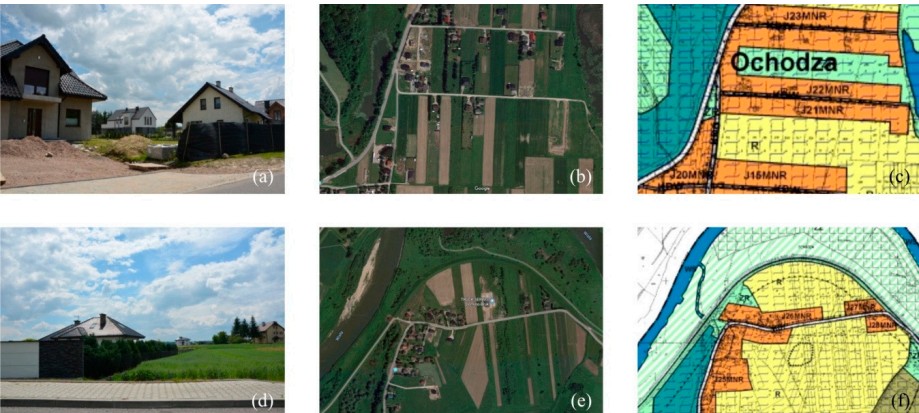

**Figure 13.** Processes related to the progressing semiurbanisation identified in the municipality of Skawina, part 1, Figures (**a**–**c**): New form of single-family housing as identified in Ochodza, Figures (**d**–**f**): Individual single-family houses in the fields in Ochodza, own work based on the 2016 local zoning plan, www.google.pl/maps.

The profile of spatial management is well illustrated by the satellite image with overlaid administrative boundaries. It shows the proportions of urbanised and open spaces in each administrative unit. Open spaces in villages neighbouring an urban area, Skawina or Kraków, or near state road 44, exhibit newly-built residential buildings integrated in the traditional elongated arrangement of agricultural fields. The further from urban centres and the road transport network, the more popular the clearly traditional layout of settlement units related to agricultural production (Figure 14; Figure 15).

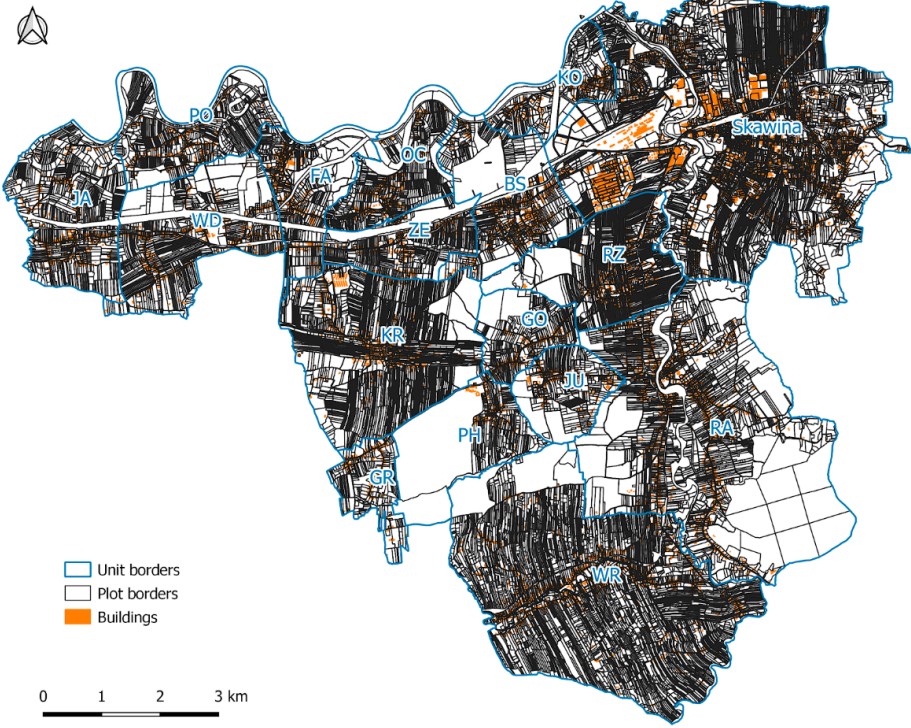

**Figure 14.** The spatial arrangement of plots of land in the municipality of Skawina.

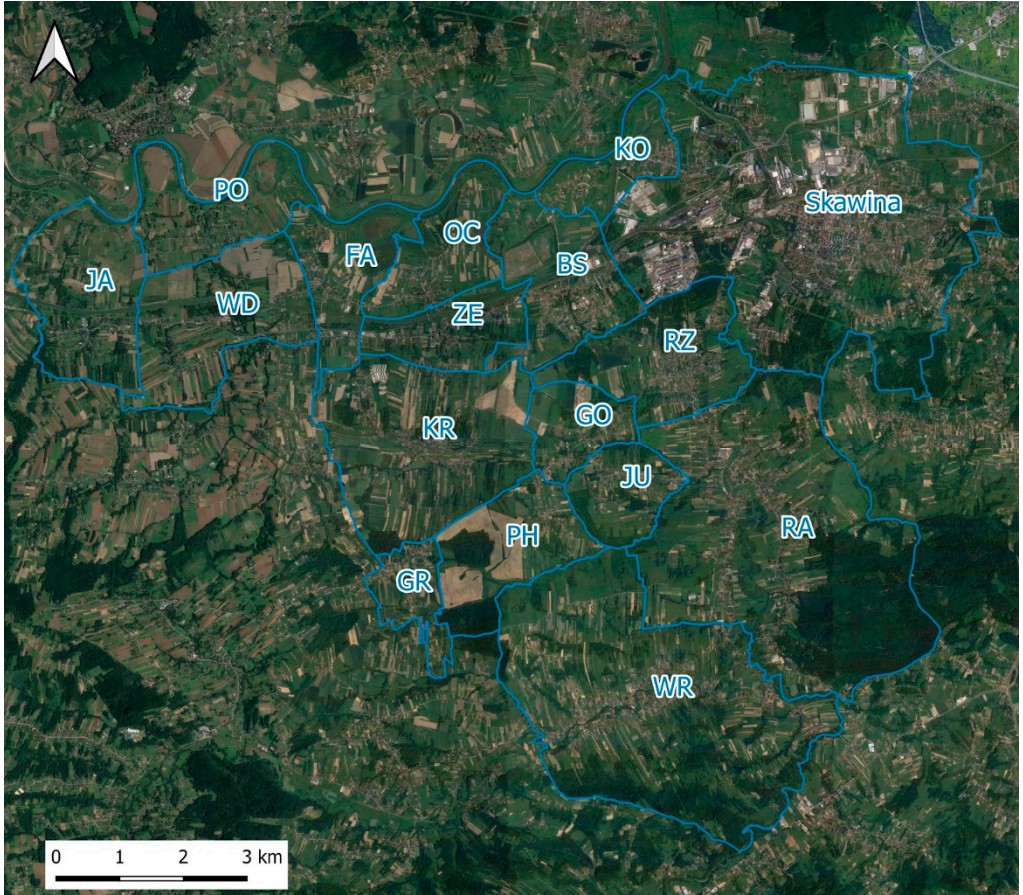

**Figure 15.** The arrangement of cadastral districts in the municipality of Skawina overlaid on a Google Satellite image.

## 4. Discussion

The authors have identified, in new and repaired buildings, construction materials and hardscaping such as fencing, sculptures, gates, entrances, or canopies that had not been used in traditional architecture. The development of plots, gardens, and entrance gardens without drawing on local architectural and construction traditions or using local trees and shrubs is clearly noticeable. It is a common phenomenon in the Polish highlands [56]. The spatial chaos is additionally escalated by ubiquitous advertisements and signboards, despite the 2015 Landscape Act, which was supposed to remove them [57]. The provisions of local zoning plans and the conversion of agricultural land followed by the subdivision into small construction plots typical of suburban municipalities lead to the sprawl into agricultural and forest land or other natural areas. The development of residential areas in the municipality entails the occurrence of agricultural enclaves surrounded by residential areas or land to be occupied by residential buildings according to the local zoning plans from 2006 and 2016. This type of land management is burdened with incorrect planning and so-called redundant suburbanisation [58]. Some buildings are constructed in locations not earmarked for building in the local zoning plan and zoning conditions and directions. As a result of suburbanisation, Skawina and many other suburban municipalities lose agricultural land and the natural habitats of many plants and animal species. The traditional cultural landscape is devastated there as well.

Similar problems have been noted in suburban zones of the largest cities in Poland. The literature includes publications concerning the transformation of rural areas situated in suburban zones of large Polish cities. All cases have much in common due to the sociocultural, economic, and historical context. Still, the changes differ, and the differences are caused by the location of the largest cities. Other dynamics can be seen in the suburban zone of Warsaw because of the metropolitan functions of

the capital. Łódź is the third-largest city in Poland, only 130 km east of Warsaw. The growth of the suburban zone of Szczecin is also specific because of its geographical location, between the border with Germany and Lake Dąbie. Other trends are apparent in the development of suburban zones in the Silesian and Dąbrowa Basin Metropolitan Area, a conurbation of fourteen cities with heavy industry and mines. The authors intended to present the characteristics of the Kraków region and Lesser Poland against this background.

The population had been concentrating in cities until the late 20th century. The highest value of urbanisation was reached in Poland in 1997 [59]. At the initial stage of urbanisation, only open spaces within city limits were developed. The socioeconomic transformation in the early 21st century advanced the dynamics of suburbanisation processes, which resulted in developments expanding outside of urban areas and aggressively invading rural zones. Today, suburbanisation is defined in Poland as the uncontrolled urbanisation of rural settlements located within convenient commuting reach. As these changes are chaotic in nature, they lead to the degradation of the value of the rural landscape (cultural heritage), the loss of cultural identity, and the loss of social bonds [14]. Processes related to suburbanisation and their adverse consequences have been noted to increase in eight metropolitan areas in Poland: Warsaw [58,60–62], Katowice Conurbation (also known as the Silesian and Dąbrowa Basin Metropolitan Area) [63,64], Gdańsk Conurbation [65,66], Poznań [33,67–69], Kraków [50,70–72], Wrocław [67,73–76], Łódź [77,78], and Szczecin [15]. The nature and dynamics of suburbanisation are different in different metropolitan areas, mostly due to the different profiles of municipalities. A particular principle has emerged, according to which built land increases in municipalities neighbouring the central city that are within 20 min of commute time. The scale and dynamics of the transformation of the suburban countryside were found to depend on the location of the municipality in relation to the central city. Municipalities directly adjacent to the central city exhibit a much higher degree of urbanisation than other rural units in the city's impact zone. A higher degree of urbanisation was noted for places that can be easily reached by road in up to 30 min. The area of land intended for development in these localities is several times larger than the area of the currently built land [15]. Suburbanisation often leads to the marginalisation of the village proper through functional, technical, and spatial degradation. Urban features have been introduced into rural areas, which is a threat to the traditional agricultural nature of the countryside. This is true not only for single-family housing but also for multi-unit buildings that serve as dormitory settlements for the central city and can be the beginning of a town or suburban city district. Apart from the destruction of rural nature, the uncontrolled growth of the suburbs results in spatial chaos and barriers for sustainable development. When attempting to counter the effects of suburbanisation, it is essential to identify processes in rural areas and suburban zones [79–83].

According to Markowski [84], suburbanisation processes that have been in place for years in European agglomerations have led to a continued transformation of cities into regions based on numerous functional and spatial relations independent of administrative boundaries. New, hitherto unidentified phenomena that cause grave spatial, social, and economic changes, threatening various forms of cultural heritage, hinder any attempt to study, classify, and delineate the problem. Nevertheless, an effort to control phenomena related to suburbanisation is vital to protect the cultural identity and environmental values, and to create proper functional, spatial, and social structures [12,61]. In the European Union, the developed rural area grew, while the rural population decreased in the period from 1960 to 2010 [85]. Studies on French metropolitan areas from 1990 to 1999 demonstrated that processes related to suburbanisation, and particularly residential developments, social segregation [86].

The present results show the strength of internal interrelations within agglomerations, their neighbouring towns, and rural areas.

Suburbanisation pressure is reflected in decision-making and development projects in villages that are part of municipalities around large cities. The decisions could contribute to the changing nature of their space. The dynamic of the transformations identified in the mixed urban and rural

municipality of Skawina is directly related to its location in the impact zone of Kraków. The results present the dynamic of changes, the main directions of the transformations leading to the urbanisation of the countryside, and their consequences for the rural landscape. This dynamic translates into the earmarking of excessive areas for residential estates in local zoning plans. This process has been noted in large cities and suburban municipalities all over Poland in recent years. When excessive land is earmarked for settlement and associated functions, the landscape grows dissonant and the natural environment is overexploited and destroyed. Suburban villages lose their environmental and cultural qualities, related to agriculture, and new features are introduced that disturb the traditional cultural landscape.

The number of buildings in villages is not decisive for the degree of changes resulting in the dissonance related to the urbanisation index. The location of the village and the development continuity along the main transport routes exhibit a much more significant impact, which is evident in KO and RZ contrasted with BS, where residential developments are separated from the residential buildings of the town by an industrial zone. Another factor indicative of urbanisation is the plot size. Smaller plots may suggest discontinued agricultural operations (and subdivision to sell) or that the owner is from an urban area and considers the countryside a "dormitory". The results indicate that the areas at the greatest risk of semiurbanisation in Skawina are the villages bordering on urban areas of both the town of Skawina: Rzozów (RZ) and the city of Kraków: Kopanka (KO), all within the Kraków Metropolitan Area.

Suburban zones of large cities exhibit a trend towards an urban-rural continuum, which is a transition zone between a city and the countryside [87]. The intensity of spatial development in the zone is related to the distance to the central city, which research [88] set to 5 km. It is apparent in the villages in the northern part of the municipality of Skawina.

The reasons seem to be easy access to urban services and short commute for residents of rural areas. Note the village of Wielkie Drogi (WD). It is located far from rural centres and borders on Jaśkowice, where the changes caused by urbanisation are the lowest. The high value of the change index might be caused by the vicinity of state road 44 from Kraków to Oświęcim. The higher change index than for Facimiech, Zelczyn, and Borek Szlachecki, situated closer to the towns of Skawina or Kraków, may result from high landscape qualities or other factors, which can be identified in significantly more detail in further research. The identification of adverse phenomena and their delineation in the immediate vicinity of rural cultural heritage assets may help create spatial policies and preserve traditional elements, their surroundings, and vistas in good condition. Such actions are of vital importance, for although it is impossible to stop the processes related to the functioning and development of cities, it is possible to protect enclaves of the rural cultural landscape.

The effort to minimise the adverse effects of suburbanisation should take different forms in such cases.

It is crucial for actions taken by municipal authorities to be tailored to the nature of the village, its critical environmental and cultural qualities, and the dynamics of changes, while remaining in line with rural development models [18,89].

What restricts the development of harmonious space in suburban villages from the spatial planning and spatial management perspective is the use of dysfunctional spatial planning tools [89–91]. First, too large areas in local zoning plans are converted into developable land for new housing schemes. The second problem is the legal possibility to build a house in locations not earmarked for residential buildings with an outline planning permission. The third problem are the permits to build structures at variance with local zoning plans regarding the nature of the building, its dimensions, roof, and colours. These problems occur in Poland in general and cause spatial chaos.

The next research steps could be social research involving a dialogue with residents and the development of a tool to provide a foundation for the protection of the traditional, regional landscape and build local awareness. An attempt could be made to sensitise new residents from urban areas to

the issues of the countryside through social dialogue, so that new developments would be in line with its profile.

Leaflets could be offered in the municipality administration centre or district administration centre with:

1. a specification for the form of new developments, a catalogue of acceptable solutions as variants of building shapes, façade materials and colour schemes, fencing, and the profile of the garden together with plant species;
2. issues concerning the functioning of the countryside and preservation of its profile;
3. questions concerning the preservation of the natural environment.

People who already live in rural areas can be encouraged to actively participate in the life of the village, events focused on traditions, or the promotion of traditional products during fairs. Relationships with original residents can gradually drive a change in the perception of rural areas and foster care for their cultural heritage and the preservation of their profile.

## 5. Conclusions

Cultural heritage is connected to agricultural sites and areas influenced by agricultural activity. It has been undergoing significant changes recently, caused by the influence of urban areas and suburbanisation. The changes affect the traditional rural landscape, included in developed areas. Dissonant objects appear in urban areas that invade the cultural landscape of rural areas. The degree and intensity of the "urban" penetration differs depending on various conditions. They can be estimated from the number and types of objects interfering with the rural landscape, for example.

It is particularly difficult to stop suburbanisation processes that interfere with the rural environment. The knowledge of the degree of landscape transformations facilitates reasonable planning policies, space development, and appropriate management.

The paper has demonstrated that rural areas in the municipality of Skawina are undergoing spatial changes resulting from both the growth of the suburban zone of Kraków and the development of the town of Skawina, which is an important industrial hub in the region, with a large job market. The overlapping impact zones of the city and the town yield a specific context for semiurbanisation processes, the results of which have been presented in detail in the paper. The villages most at risk of transformations related to urbanisation, which are those in need of preventive actions against the adverse effects of suburbanisation, are the villages neighbouring urban areas, administrative centres of mixed urban and rural municipalities, and large cities with good road transport. The pace of semiurbanisation changes is much slower in villages far from dominant urban areas and transport infrastructure.

**Author Contributions:** Conceptualization, B.O. and M.W.-M.; methodology, M.W.-M., B.O., B.P.; software, B.P.; validation, M.W.-M., B.O. and B.P.; formal analysis, B.O., M.W.-M.; investigation, B.O., M.W.-M.; resources, B.P., B.O.; data curation, B.O., B.P., M.W.-M.; writing—original draft preparation, M.W.-M., B.O.; writing—review and editing, B.P., M.W.-M.; visualization, B.P., B.O.; supervision, B.O., M.W.-M.; project administration, M.W.-M., B.O.; funding acquisition, B.P. All authors have read and agreed to the published version of the manuscript.

**Funding:** This research was funded by the project 'Cultural heritage of small homelands' No. PPI/APM/2018/1/00010/U/001, financed by the Polish National Agency for Academic Exchange as a part of the International Academic Partnerships, funded with a subsidy of the Ministry of Science and Higher Education for the University of Agriculture in Kraków for 2020; Civil engineering and transport 030008-D018/KGPiAK/2020 Environmental engineering, mining, and energy science 030008-D014/KGPiAK/2020.

**Acknowledgments:** The authors would like to thank the anonymous reviewers for their thorough work with the manuscript and for providing constructive and insightful comments on this paper.

**Conflicts of Interest:** "The authors declare no conflict of interest." The funders had no role in the design of the study; in the collection, analyses, or interpretation of data; in the writing of the manuscript, or in the decision to publish the results.

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
