# Peer review of "Urban Features in Rural Landscape: A Case Study of the Municipality of Skawina"

_sustainability, doi:10.3390/su12114638_

Round 1

Reviewer 1 Report

The paper “Urban features in rural landscape. A case study of the Municipality of Skawina” focuses on the urban sprawl phenomenon in Poland. Although it relies on a very detailed analysis, the paper has several shortcomings that should be addressed before being published.   

Introduction: Lines 37-43: the statements provided by the authors need to be referenced.

Lines 45-46: the authors write: “In Norway, agriculture is seen both as a threat to and a caretaker of cultural heritage”. It applies in several areas, I guess…

In general terms, authors should better specify the aims and focus of their work.

Methodology: it is very weak. Authors refer to three stages, but no details are provided to the reader on the methods adopted.

Lines 129-132: these sentences are out of context here.

Results and discussion: part of the discussion can be found into the conclusion section. I strongly suggest moving the text now included as conclusion into the discussion section. Moreover, lines 348-366 should be considered as literature review or integral part of the introduction.

Conclusion should be rewritten.

Author Response

Dear Sir or Madam,

first of all, thank you very much for your valuable comments which we believe helped us to improve the quality of the paper. We did our best to include all the requests with respect to the recommendations of both reviewers. The specific identification of the changes made in the paper are written below each bullet of your review (red colour)-Please see the attachment.

Kind regards

Authors

Review Report 1

1.              Introduction:Lines 37-43: the statements provided by the authors need to be referenced

Re: The Reviewer's comments have been taken into consideration. The number of citations has increased (we would like the reviewer to recommend some other if needed).

The following literature has been cited:

-     Vizzari, M., Peri-Urban Transformations in Agricultural Landscapes of Perugia, Italy,  Journal of Geographic Information System, 2011, 3, 145-152

-     BaÅ„ski, J.,WesoÅ‚owska, M., Transformations in housing construction in rural areas of Poland's Lublin region—Influence on the spatial settlement structure and landscape aesthetics, Landscape and Urban Planning, 2010, 94, 116-126.

-      Cabana, D,. Ryfield, F, Crowe, Tasman, P.; et al. Evaluating and communicating cultural ecosystem services, Ecosystem Service 2020,  42.  

-         Bui, Huong T.; Jones, Thomas E.; Weaver, David B.; et al. The adaptive resilience of living cultural heritage in a tourism destination,  Journal of Sustainable Tourism, 2020, 28, 1022-1040   

  1. Lines 45-46: the authors write: “In Norway, agriculture is seen both as a threat to and a caretaker of cultural heritage”. It applies in several areas, I guess

Re: According to this useful suggestion, we have added the text: 

In many countries (such as Norway), agriculture is seen both as a threat (often large-area, mechanised agriculture) to and a caretaker (mostly traditional crops) of cultural heritage [8]. Cultural heritage is the sum of what nature provides, such as areas with valuable environment or used for farming, and anthropogenic contribution, such as settlement arrangements or traditional development typical of the region and determined by the local lifestyle [9]. 

  1. In general terms, authors should better specify the aims and focus of their work.

Re: According to this useful suggestion, the content of the Introduction was expanded:

The first step to determine the transformations in the municipality of Skawina caused by the growth of Kraków and its suburban zone was to investigate the internal conditions in the municipality such as its spatial development or functional structure and external conditions, demographic transformations in particular. Next, the development of the settlement structure over the recent years and land management changes were investigated. The primary objective of the paper was to analyse typical urban features and functions introduced into rural areas and identify the transformations and dynamics of the changes. The data can be used to build schemes to prevent adverse effects of semiurbanisation of rural areas in the municipality of Skawina.  

  1. Methodology: it is very weak. Authors refer to three stages, but no details are provided to the reader on the methods adopted.

Re: The research methods have been changed and described in detail following the comment. The text has been introduced to section 2. 1. Research method

Suburbanisation is manifested not only through urban features appearing in the rural landscape but also in a broader spatial context. This is why the research was divided into three stages and problem perspectives. 

Stage I – the spatial analysis method – geoprocessing methods

- calculation of the number of residential buildings in the investigated area (with a DTO10k layer)

- analysis of the spatial distribution of buildings in localities – DTO10k layers

- analysis of land cover forms as classified in the DTO10k, particular focus on developed and urbanised areas

- determination of differences in areas of developed land based on spatial layers of the local zoning plans for 2006, 2014, and 2016

Stage II – identification of individual examples of items at variance with rural cultural heritage – in situ investigation, field survey, including inventory with photographs, and drawings

Stage III – space diagnosis based on an original method for assessing the degree of interference with the original rural fabric supplemented with an analysis of the spatial structure of plots (data from the District Land Surveying and Cartographic Documentation Centre) arranged in traditional elongated patterns and analysis of satellite images (LiDAR data).

The major identified settlement network transformations related to demographic changes in the municipality of Skawina included intensified development with new single-family houses of urban nature. The analysis involved features alien to the rural landscape, including street furniture, hardscape, façade colours contrasting with the landscape, modern gardens, forms of buildings at variance with rural traditions, and modular fences.

Tab. 1. Results of the field survey – investigation of features alien to the rural landscape.

No. of single-family houses

Street furniture/hardscape alien to countryside

Façade colours contrasting with landscape

Modern gardens, typical of terraced housing: lawn + hedge

Building form at variance with rural tradition

Modular, non-traditional fencing

Synthetic index

V

NV

V

NV

V

NV

V

NV

V

NV

V

1

BS

392

0,01

0,4

0,03

0,6

0,3

1,6

0,3

1,1

0,3

2,1

5,7

2

FA

180

0,01

0,4

0,05

0,9

0,4

2,2

0,4

1,4

0,03

0,2

5,1

3

GO

151

0,01

0,4

0,5

0,9

0,1

0,5

0,3

1,1

0,1

0,7

3,6

4

GR

104

0,01

0,4

0,05

0,9

0,1

0,5

0,4

1,4

0,05

0,4

3,6

5

JA

298

0,01

0,4

0,01

0,2

0,5

0,3

0,1

0,4

0,05

0,4

1,5

6

JU

163

0,01

0,4

0,03

0,6

0,2

1,1

0,3

1,1

0,01

0,1

3,1

7

KO

201

0,1

3,6

0,1

1,8

0,3

1,6

0,7

2,5

0,4

2,8

12,4

8

KR

391

0,3

1,1

0,1

1,8

0,3

1,6

0,2

0,7

0,1

0,7

6,0

9

OC

239

0,01

0,4

0,05

0,9

0,1

0,5

Less than 0.1

0,2

0,02

0,1

2,1

10

PH

120

0,01

0,4

0,01

0,2

Less than 0.1

0,3

Less than 0.1

0,2

0,02

0,1

1,1

11

PO

175

0,05

1,8

0,03

0,6

0,05

0,3

0,2

0,7

0,8

0,6

3,9

12

RA

965

0,03

1,1

0,6

1,1

0,1

0,5

0,4

1,4

0,1

0,7

4,9

13

RZ

412

0,03

1,1

0,05

0,9

0,3

1,6

0,3

1,1

0,4

2,8

7,5

14

WD

299

0,05

1,8

0,1

1,8

0,3

1,6

0,4

1,4

0,4

2,8

9,5

15

WR

758

0,02

0,7

0,05

0,9

0,1

0,5

0,2

0,7

0,1

0,7

3,6

16

ZE

254

0,05

1,8

0,1

1,8

0,2

1,1

0,2

0,7

0,1

0,7

6,2

V - Value

NV – Normalised Value

Values in Table 1 present results of the field survey. The original values were noted as a percentage of alien forms in the total number of buildings in the localities. The normalised value was calculated by expressing a variable as a multiplicity of the mean, using Formula (1):

         (1)  

where: xi’ – the normalised value

                xi – the value calculated based on the field survey

                xav – the mean value of variable x

The synthetic index summarises normalised values of variables, calculated based on Formula (2): 

         (2)

The units (localities) were classified using the Jenks natural breaks method. Four classes were identified based on the values of the synthetic index. High values of the index reflect significant transformation / occurrence of alien forms in the landscape, while low values mean low level of transformation in the units (localities).

  1. Lines 129-132: these sentences are out of context here.

The conclusions may support future actions to prevent any adverse consequences of suburbanisation in the suburban zones of Kraków and the semiurbanisation it entails. They may be the foundation for determining conditions for the protection of local values, regional architecture, culture, and landscape as well. 

Re: This comment made the authors realise the sentence was just a statement, not a conclusion of the paper and should be removed.

  1. Results and discussion: part of the discussion can be found into the conclusion section. I strongly suggest moving the text now included as conclusion into the discussion section. Moreover, lines 348-366 should be considered as literature review or integral part of the introduction

Re: Lines 348–366 have been moved as suggested.

  1. Conclusion should be rewritten.

Re: The Conclusions have been rewritten as advised. Part of the text has been moved to section 3. Results and discussion, the other part was slightly expanded.

Cultural heritage is connected to agricultural sites and areas influenced by agricultural activity. Cultural heritage has been undergoing significant changes recently caused by the influence of urban areas and suburbanisation. The changes affect the traditional rural landscape, including developed areas. Dissonant objects appear in rural areas that invade the cultural landscape of rural areas. The degree and intensity of the ‘urban’ penetration differ depending on conditions. They can be estimated from the number and types of objects interfering with the rural landscape, for example.

. It is particularly difficult to stop suburbanisation processes that interfere with the rural environment. The knowledge of the degree of landscape transformations facilitates reasonable planning policies, space development, and appropriate management.

The paper has demonstrated that rural areas in the municipality of Skawina are undergoing spatial changes resulting from both the growth of the suburban zone of Kraków and the development of the town of Skawina, which is an important industrial hub in the region with a large job market. The overlapping impact zones of a city, Kraków, and town, Skawina, yield a specific context for semiurbanisation processes, the results of which have been presented in detail in the paper. The villages most at risk of transformations related to urbanisation, which are those in need of preventive actions against adverse effects of suburbanisation are the villages neighbouring on urban areas – administrative centres of mixed urban and rural municipalities – and bordering on large cities with good road transport. The pace of semiurbanisation changes is much slower in villages far from dominant urban areas and transport infrastructure. 

Reviewer 2 Report

Review of “Urban features in rural landscape. A case study of the Municipality of Skawina.”

This paper examines suburbanization in the rural landscape around Skawina, a municipality outside of Krakow in Poland.

The analysis demonstrates changes that have taken place in the landscape, and the ways in which suburbanization is occurring with suburban housing being built and population growing. The paper suggests that there is a potential for conflict over urban versus rural uses.

It is an interesting enough topic but I am left with a feeling of “so what?” What does this paper tell us about suburbanization more broadly or what does it tell us about the reasons for this occurring in the context of Poland and the Krakow region? My biggest concern about this paper is that it does not really engage with the literature in ways that deal with the “so what” factor. The author(s) provide nice maps and figures to show how suburbanization is happening but they offer little in the way of demonstrating how this process of suburbanization is different or similar to processes of suburbanization explored in the literature. There is a tremendous amount of literature on suburban sprawl in the United States, for example, and so it might be interesting for the author(s) to consider how suburbanization in rural areas in the context of Poland might be different or similar to the process in the U.S. In the same way, there is recent literature about suburbanization processes in post-communist countries that might be relevant. I would suggest the author(s) examine the recent book Suburban Governance: A Global View for some comparisons. This book also references lots of interesting literature that could be useful. Basically, the author(s) need to put their work in some sort of context.

In addition and related, the author(s) do not really engage with any theory. Why do they think this suburbanization is happening? What are the drivers? Are their larger global processes at work? How is planning or policy at the local or regional level affecting these changes? Is there something unique about this policy environment that we should know about? Or is it similar to what you might see in other regions both in Poland and elsewhere? 

Ultimately, without engagement with theory and with the literature on suburbanization and suburban sprawl, this paper reads more like a report rather than a quality academic journal article.

As far as the methods, it is not completely clear exactly how the data was analyzed. It seems like it’s a combination of LiDAR and zoning data. Is that correct? I think a separate section on the actual datasets used in the study and how they were analyzed would be useful.

One question I had was whether or not some of the suburban housing being built was for second homes that might belong to families living in Krakow. Is Skawina and area that residents of Krakow go to “get away” for the weekend?

Figure 4 and Figure 5: the legends should be translated into English

Author Response

Dear Sir or Madam,

first of all, thank you very much for your valuable comments which we believe helped us to improve the quality of the paper. We did our best to include all the requests with respect to the recommendations of both reviewers. The specific identification of the changes made in the paper are written below each bullet of your review (red colour)-Please see the attachment.

Kind regards

Authors

Review Report 2

  1. It is an interesting enough topic but I am left with a feeling of “so what?” What does this paper tell us about suburbanization more broadly or what does it tell us about the reasons for this occurring in the context of Poland and the Krakow region? My biggest concern about this paper is that it does not really engage with the literature in ways that deal with the “so what” factor. The author(s) provide nice maps and figures to show how suburbanization is happening but they offer little in the way of demonstrating how this process of suburbanization is different or similar to processes of suburbanization explored in the literature. There is a tremendous amount of literature on suburban sprawl in the United States, for example, and so it might be interesting for the author(s) to consider how suburbanization in rural areas in the context of Poland might be different or similar to the process in the U.S. In the same way, there is recent literature about suburbanization processes in post-communist countries that might be relevant. I would suggest the author(s) examine the recent book Suburban Governance: A Global View for some comparisons. This book also references lots of interesting literature that could be useful. Basically, the author(s) need to put their work in some sort of context. In addition and related, the author(s) do not really engage with any theory. Why do they think this suburbanization is happening? What are the drivers? Are their larger global processes at work? How is planning or policy at the local or regional level affecting these changes? Is there something unique about this policy environment that we should know about? Or is it similar to what you might see in other regions both in Poland and elsewhere?

Re: Thank you for your valuable comment. The objective of the paper was to describe transformations in the municipality of Skawina caused by the growth of Kraków and its suburbs with a particular focus on alien features introduced into the traditional rural landscape. Sixteen localities in the municipality of Skawina were analysed. It was not the goal of the paper to investigate drivers of suburbanisation. The authors are aware that suburbanisation processes differ by country, have different backgrounds, causes, and historical motivations. Nevertheless, the authors took the comment into account and cited the proposed literature in the context of a post-socialist country.

The following text was added to the Introduction: 

It was not possible to make long-term and consistent plans for rural areas because of many agrarian reforms motivated politically in Poland; there were nine major agrarian reforms in the period of 60 years from 1933 to 1999. Political transformations towards a free-market economy after 1989 were reflected in exceptionally dynamic changes in the spatial and functional structure of rural settlements. As a consequence, the social and economic structure of the then-agricultural village changed. The concepts for rural urbanisation strove mostly to present it as an overwhelming process of changes that replaces old structures. The idea of urbanisation was, to a large extent, a negation of rurality understood as a particular cultural tradition, sense of identity, social, cultural, economic, and landscape otherness [17]. The modern rural layout is considered a vitally important component of the heritage and thus local resources of the village. It is defined by the law as a ‘rural planning scheme with building complexes, individual buildings, and landscaped green areas arranged according to historical ownership and functional structure, including that of streets or road network’ [19].

The suburbs have developed differently in different countries. Many researchers [20-22] have confirmed that trends in post-communist countries are similar to those in developed, capitalist countries, but slowed due to the nature of their socioeconomic growth. The countries have specific historical factors, geographical conditions, housing stock, economic processes, or urban-to-rural migration [23,24]. Because of centralised socialist planning, urbanisation in Poland had a strikingly different nature than in capitalist cities. Most urban regions were monocentric: from the countryside (the provider for the city) towards the core of the city where jobs and services were concentrated.  The sociopolitical transformation introduced a market economy. This socioeconomic reorganisation has been reflected in the transformation of the landscape of former socialist cities and their surroundings. Suburbanisation, with its spatial dispersion, has become the dominant mode of city growth. New commercial and industrial zones were established as it progressed. New project schemes paved the way for new suburban job hubs [25]. The suburban growth phase was bound up with the size of the city and the level of economic development. The influx of the urban population to suburbs started first in centres that completed the transitional period [26].

An important factor affecting the economic growth of the town of Skawina was the establishment of an aluminium smelter after World War II. Other investments followed: a power plant and new housing developments. The population grew as the nearby plants promised job opportunities. The trend continues today. Agriculture is slowly dying; arable land is often set aside or transformed into fallow land (Monograph of Skawina, 2014). Residents have mostly abandoned agriculture in favour of jobs offered by numerous businesses in the area.

  1. Ultimately, without engagement with theory and with the literature on suburbanization and suburban sprawl, this paper reads more like a report rather than a quality academic journal article

Re: As suggested, the paper now includes text about suburbanisation, including a general description of the situation in postsocialist countries such as Poland.  

  1. As far as the methods, it is not completely clear exactly how the data was analyzed. It seems like it’s a combination of LiDAR and zoning data. Is that correct? I think a separate section on the actual datasets used in the study and how they were analyzed would be useful.

Re: Thank you for your valuable comment. It is similar to a comment by Reviewer 1. It has been taken into consideration by adding relevant text to section 2. 1. Research method, please see the text in red.

  1. One question I had was whether or not some of the suburban housing being built was for second homes that might belong to families living in Krakow. Is Skawina and area that residents of Krakow go to “get away” for the weekend?

Re: Most houses built in the suburbs of Skawina are regular homes, not second homes.

  1. Figure 4 and Figure 5: the legends should be translated into English

Re: The comment has been taken into consideration. The legend has been translated into English.

Reviewer 3 Report

This is an interesting article and has relevance for planners, cultural historians, rural sociologists, and other

This is an interesting article and has relevance for planners, cultural historians, rural sociologists, and others interested in changes to rural landscapes.  It is in need of major revision throughout the paper before it can be considered for publication.  More specifically,

  1. The entire paper needs editing and proofreading by someone fluent in English.  There is a lot of repetition throughout the paper, and places where main ideas are unclear.  
  2. The abstract needs to be rewritten to more succinctly specify the research questions, findings, and implications.
  3. Emphasize more strongly that this is a case study and discuss how case studies can add to our understanding of this particular situation and to other similar situations.
  4. The literature review needs to be strengthened. Consider the body of literature on the rural-urban interface or rural-urban fringe, and perhaps other areas such as migration and suburbanization. 
  5. Regarding the region of the study, add more information on the history, economy, and traditional heritage of the area to provide greater context for the reader.
  6. Discuss the connection between cultural heritage and architecture/economy/agriculture in more detail. Be clearer about why the loss of this heritage is important to be concerned about.  Include information why it is important not just for Poland but for other places as well.
  7. I would be interested in knowing more about the influences of other factors in changing the landscapes besides just outmigration. What is the role of the media, changing income levels, access to land and credit, the influence of housing developers, etc.? 
  8. Clearly specify the research questions of the study.
  9. Some of the text describing the local area should be moved to a section describing the area so that there is a clearer separation with the study results.
  10. Carefully proofread so that graphs are in English. Also provide an English translation to the Polish sources cited.
  11. Provide a greater discussion of ‘spatial dissonance’ and ‘spatial chaos.’
  12. Include a section on implications for planners and communities.
  13. Discuss limitations to the study and next steps for research.
  14. Link the findings back to the literature review.

Round 2

Reviewer 1 Report

This new version of the paper has been significantly improved.

Nevertheless, it is in need of minor revisions throughout the paper before it can be considered for publication. 

More specifically, I would suggest having results and discussion sections separated.

The results section needs a clearer explanation of how the different stages/steps of the methodology have added information and enriched the analysis. The reader get lost in following the text without a clear understanding of the research process. For example, starting from line 294 until line 323, are you talking about the land use changes from different perspectives? Which is the relationship between the 13.58% increase of urbanized area with the urban plans? It needs a clearer explanation. In general, the text needs to be much more focused and less redundant.

Reviewer 2 Report

The authors have improved the paper. My only suggestion would be to write out the different stages of the methodology rather than putting just bullet points or you could put this in table format and then have text to explain each bullet in the text. It just looks strange to see bullet points listed this way in the methods section.

Reviewer 3 Report

There have been a number of improvements to the paper and it is more interesting to read.  Substantial revisions will make it a stronger paper with greater relevance theoretically and with practical implications.

  1. The wording throughout the paper could be improved for readability.  Some of the sentences are quite lengthy and choppy.  Write smoother sentences and reduce information in clauses separated by commas and in parentheses wherever possible. 
  2. It would be helpful to have a native English speaker or teacher review the paper. There are places where the wording reads very generally and is not at an academic level. 
  3. Some reordering in the introduction would be helpful to make a stronger justification for the study. Why does it matter if there is a loss of cultural heritage?  Are there debates about this in the literature, with others such as government officials seeing it as a positive trend?  The information by KamiÅ„ski should be placed earlier in the introduction, as well information toward the end of the paper.
  4. I thought the information about socialist planning was really interesting and provided some of the background to the region that I was looking for.
  5. I would like to like to have a clearer understanding of what the traditional Polish rural settlement was like. Did these settlements have shops, a church, a school, homes, etc.?   What kind of agriculture?  (the original rural fabric as mentioned on line 208).  Provide a brief description.
  6. What characterizes an urban/suburban settlement? (there is mention of characteristics in the methods section, but it would be helpful to have a discussion of these things earlier in the paper).
  7. Summarize more succinctly the patterns of suburban growth found in other places and why it is a concern for rural places.
  8. The analysis needs to flow from the literature review. Discuss residential buildings, land cover, etc. in the introduction as part of the discussion of urbanization/suburbanization.
  9. Clearly state the research questions.
  10. Methods – this section needs an introduction and reordering. Describe the methods/stages rather than list as bullet points.  Why the years of 2006, 2014 and 2016?  Explain what DTO 10K layer is.
  11. The detailed description of the region needs to be placed earlier in the paper.
  12. Provide more explanation about the development of the index.
  13. I really enjoyed seeing the photos.
  14. Tie the literature review and discussion together. How does this study increase our understanding of rural places?
  15. The paper should move beyond a study of this region to include implications for other regions.
  16. Include a discussion of limitations and next steps for research.

Round 3

Reviewer 3 Report

There have been a number of improvements to the paper and it is more interesting and enjoyable to read.  The addition of Figure 1 is helpful in understanding what the authors have done for their research.  There is a nice description of the location of the study which helps the reader understand and visualize past and current conditions.  The readability has improved considerably. There is still editing needed, but I assume the editors of the journal will be able to assist with this. 

I have some suggestions that will continue to improve the paper:

  1. It would be helpful to have the purpose of the paper stated earlier in the paper, near line 203, then introduce the information about Skawina.
  2. Align the findings to the three stages. The findings seem to jump around between the stages.  Labelling the headings with the stage number and a description would be helpful. 
  3. Any descriptive information about Skawina or Krakow that is not related to the findings should be moved to the description of Skawina earlier in the paper. For example, the rational for selecting the region is presented in lines 381-386.  This information should be mentioned earlier.  Similarly, background information on permits from the 1990’s and other background information from lines 425-441 and Lines 511-515 should be moved to an earlier section and brought back into the discussion section, unless clearly a finding from the research. 
  4. Eliminate any redundant information.
  5. I appreciated the suggestion about social research and dialogue with residents. Are there ways new residents could be connected to the traditional landscapes on which they now live?
